# Model Architecture Controls Gradient Descent Dynamics: A Combinatorial Path-Based Formula

## Abstract

Recently, there has been a growing interest in automatically exploring neural network architecture design space with the goal of finding an architecture that improves performance (characterized as improved accuracy, speed of training, or resource requirements). However, our theoretical understanding of how model architecture affects performance or accuracy is limited. In this paper, we study the impact of model architecture on the speed of training in the context of gradient descent optimization. We use the ideas from prior work that shows gradient descent can be modeled as a first-order ODE and use ODE's coefficient matrix $H$ to characterize the convergence rate. We introduce a simple analysis technique that enumerates $H$ in terms of all possible "paths" in the network. We show that changes in model architecture parameters reflect as changes in the number of paths and the properties of each path, which jointly control the speed of convergence. We believe our analysis technique is useful in reasoning about more complex model architecture modifications.

## 1 Introduction

Gradient descent and its variants are the cornerstones of deep learning. The theoretical properties of gradient descent have been widely studied in the literature; study of the convergence bounds and guarantees [6, 12, 15, 13], the characterization of the local geometry of stationary points [21, 11, 10, 19, 25, 30, 24, 23, 13], exploration of better algorithms to optimize the descent process [28, 16, 26, 33, 7] are just a few example of active research areas in this domain. Another major research area is the exploration of network architecture and its impact on performance. In recent years, network architecture search and design have shown major performance boosts in various aspects of deep learning [36, 35, 18, 8, 34].

Recent theoretical results are trying to relate model architecture and gradient descent properties. It has been shown that over-parametrization in width guarantees convergence in deep neural network [15]. It has also been shown that increasing depth has an impact similar to adding momentum optimization and adaptive learning rate to the objective function [4].

The gradient descent process can be described by a system of first-order differential equations in the continuous-time limit, in the form of $\dot{\mathbf{x}}(t) = H(t)\mathbf{x}(t)$. In this paper, we study the properties of the coefficient matrix $H(t)$, which governs the dynamics of the gradient descent. Specifically, we formulate $H(t)$ in terms of all possible paths in the network. This representation is powerful as it enables us to analyze the gradient behavior symbolically and abstracts away the unnecessary bookkeeping of derivatives and weights. Specifically, we use this representation to explore the followings. (1) We study the impact of width on convergence. Using a simple path counting argument, we show that $H(t)$ is a sum of $m$ i.i.d. terms where $m$ is dictated by the width of the network. This has been implied in the work of Simon Du, et al. [12] for the special case of a 2-layer RELU-activated network. We contrast the ease of arriving at such conclusions using our representation against methods used in previous work [12, 15, 13, 6]. (2) We study the impact of depth on convergence. While prior work [3] established the relationship between momentum and depth, we show a direct relationship between depth-induced momentum and the scale of the output value. We also discuss how depth-induced momentum is different than explicit momentum.

(3) We argue why the number of paths is more important than the number of nodes in a network.

## 2 RELATED WORK

Related work can be categorized into three groups which we will explain in detail below. In general, the common theme across previous works is to show that if some conditions, such as bound on models' width or depth or assumptions on geometrical properties of the loss landscape are met, gradient descent or its variance converges to a global minimum.

**Landscape Analysis** Researchers have extensively studied the properties of error surface in linear neural network ([10, 21]) and have proved convergence to a global minimum for non-convex optimizations if error surface shows certain properties, such as all saddle points to be strict (i.e. there exists a negative curvature)([11, 19, 25, 30, 24, 23, 13]). However, it has been shown that the strict saddle property is not guaranteed for deep neural networks.

**Width Analysis** A number of recent work have characterized minimum width that guarantees convergence in terms of model architecture parameters and input data. We on the other hand study how convergence rate varies with width. Examples of finding width bounds include [15] where it has been shown that for a 2-layer ReLU-activated neural network with squared loss, as long as every hidden layer is wide enough, gradient descent converges to a global minimum at a linear rate. In [14, 1], it is extended to fully connected linear networks and ResNets. Du et. al. [12] has also generalized this work to an L-layer fully connected linear neural network and shows the convergence rate as a function of depth, output dimension and least eigenvalue of the Gram matrix for large enough hidden layers. Ardalani et. al. [2] have shown empirically that for a wide range of RNN applications, increasing width will reduce the number of steps to minimum validation loss. Neural tangent kernel [22] studies convergence in the infinite width limit.

**Depth Analysis** Bartlett et al. [6] have proven for a deep linear neural network with isotropic input and identity initialization that the number of steps to $\epsilon$-proximity of the best answer is polynomial in the number of layers. Neural ODE studies the convergence in the infinite depth limit [27, 9, 20, 17]. Work that motivated our depth analysis are [3, 4]. They have shown that over-parametrization accelerate training with these extra restrictions: (1) linear neural network with (2) fully-connected layers and (3) "balanced" initialization where (4) dimensions of hidden layers are at least the minimum of the input and output dimensions. We contribute in the following ways. We characterize depth-induced momentum in value space with an inequality and compare with explicit gradient descent momentum. We observe that momentum is directly impacted by value of the sub-network. We lift the structure and activation restrictions.

## 3 NOTATIONS AND CONVENTIONS

Matrix entries are denoted with row and column indices, for instance $H = (H_{ij})$ is a matrix with entries $H_{ij}$. When $H$ is diagonalizable, we denote by $\lambda_{min}(H)$ its smallest eigenvalue. For a matrix $W$, $W(i,:)$ and $W(:,j)$ denote the row vector in the $i$-th row and the column vector in the $j$-th column, respectively. Write $\sum_{w \in W}$ for summing over all elements of $W$. We use pairs $\{X_i, y_i\}, 1 \le i \le N$ to represent labeled data where $X_i$ is input and $y_i$ is the output. We write $x_{i,k}$ for the $k$-th component of $X_i$. We denote loss by $L$. We work with feed-forward neural networks, which we sometimes denote abstractly by a function $f$, or $f(w,x)$ as a function of input $x$ and weights $w$. We denote network weights individually by $w_i$. Predictions are denoted by $u_i = f(w, X_i)$ $1 \le i \le N$. The prediction on all inputs is denoted by the vector $\mathbf{u}$. $l_p$ loss is defined as $\sum (y - u_i)^p$ for even integers $p \ge 2$. We use angle brackets to denote inner products, i.e. $\langle \overrightarrow{v}, \overrightarrow{w} \rangle = \sum_i v_i w_i$ represents inner product of two vectors. i.i.d. random variables stands for independent and identically distributed random variables. ODE stands for ordinary differential equations. NN stands for neural networks. GD stands for gradient descent.

## 4 BACKGROUND

We build upon prior work [15, 12, 14, 3, 31] which take gradient descent to a continuous time limit and derive the Gram-matrix $H$ governing gradient descent dynamics. In this section, we give a quick derivation for Equation (4) for completeness.

Convergence rate can be characterized as the rate of change in loss during the training process. If steps are infinitesimally small, the rate of change in loss can be characterized as:

$$\frac{dL}{dt} = \sum_{i=1}^{N} \frac{\partial L}{\partial u_i} \cdot \frac{du_i}{dt} \tag{1}$$

Where $u_i(t) = f(w(t), X_i)$ is the prediction on input $X_i$ at time $t$. For simplicity we focus on the dynamics of $\frac{du_i}{dt}$, which can be expanded as follows:

$$\frac{du_i}{dt} = \sum_{r=1}^{m} \frac{\partial u_i}{\partial w_r} \cdot \frac{dw_r}{dt} = \sum_{r=1}^{m} \left( \frac{\partial u_i}{\partial w_r} \sum_{j=1}^{N} -\eta \frac{\partial L}{\partial u_j} \cdot \frac{\partial u_j}{\partial w_r} \right)$$

$$= -\eta \sum_{j=1}^{N} \sum_{r=1}^{m} \left( \frac{\partial u_i}{\partial w_r} \frac{\partial u_j}{\partial w_r} \cdot \frac{\partial L}{\partial u_j} \right) = -\eta \sum_{j=1}^{N} \left( \sum_{r=1}^{m} \frac{\partial u_i}{\partial w_r} \frac{\partial u_j}{\partial w_r} \right) \cdot \frac{\partial L}{\partial u_j} \tag{2}$$

We use the definition of gradient descent and chain rule to derive the equation above. Gradient descent at infinitesimal small step is defined by:

$$\frac{dw_i}{dt} = -\eta \frac{\partial L}{\partial w_i} = -\eta \sum_{j=1}^{N} \frac{\partial L}{\partial u_j} \cdot \frac{\partial u_j}{\partial w_i} \tag{3}$$

Observe that $\overrightarrow{\frac{\partial L}{\partial u}}$ is a column vector. Hence, if we define $H_{ij}$ as:

$$H_{ij} = \sum_{r=1}^{m} \frac{\partial u_i}{\partial w_r} \frac{\partial u_j}{\partial w_r} \tag{4}$$

Then we can simplify equation (2) in a vector form as:

$$\begin{pmatrix} \frac{du_1}{dt} \\ \vdots \\ \frac{du_N}{dt} \end{pmatrix} = -\eta H \cdot \begin{pmatrix} \frac{\partial L}{\partial u_1} \\ \vdots \\ \frac{\partial L}{\partial u_N} \end{pmatrix} \tag{5}$$

Assuming $L$ is $l_2$ loss, equation (5) simplifies to the following. This result will hold up to a constant for other $l_p$ losses (see Appendix A).

$$\frac{du}{dt} = \eta H(y - u) \tag{6}$$

$$\frac{d(y - u)}{dt} = -\frac{du}{dt} = -\eta H \cdot (y - u) \tag{7}$$

Notice that this puts $\mathbf{y} - \mathbf{u}$ in a system of differential equations whose single variable analogue is:

$$\frac{df(t)}{dt} = h f(t) \tag{8}$$

The above equation's solution is:

$$f(t) = c_0 e^{ht} = c_0 (1 - \xi)^t \tag{9}$$

when $h < 0$, $e^h < 1$ and let $e^h = 1 - \xi$. $\xi$ governs the convergence rate in Equation (9). In the actual system of equations (7), $\xi$ will be determined by the minimum eigenvalue of $H(t)$, denoted $\lambda_{min}(H(t))$.

## 5 MAIN IDEA: CHARACTERIZING H IN TERMS OF NETWORK PATHS

In order to state the path decomposition formula, we first establish some notations.

- We denote the neural network abstractly with a function $f(X)$.

- For any neural network, we view the underlying network as a **directed graph** where edges are in the backward propagation direction.

- We define **path** $p$, as a series of connected nodes and directed edges. We define **an output path** as a path that starts with the output node. We denote the length of $p$ by $l(p)$.

- We **index** the nodes as follows. Index the output node by 0, the one after by 1, and so on. We denote the $j$-th node in the $i$-th path by $p_j^i$.

- We allow different activation functions at nodes. We denote the **activation function** at $p_s$ by $\sigma_s$.

- An example path that ends in an input node may look like this:

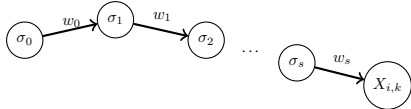

Figure 1: Illustration of a multilayer perceptron

- The **post-subnetwork at $\mathbf{w_s}$** is defined as the network formed by the subgraph rooted at the right vertex of $w_s$. The **pre-subnetwork at $\mathbf{w_s}$** is defined as the union of all output paths ending in $w_s$, denoted by $pre(w_s)$.

- For every input $X_i$, we denote the **activation** value at the node after $w_s$ with $\mathbf{f}[\mathbf{w_s}](\mathbf{X_i})$. Note that this is equivalent to the value at the right vertex of $w_s$ once $X_i$ propagated through the post-subnetwork at $w_s$.

- We denote the local gradient wrt. input at node $s$ along the path $p$ with $\tau_{p_s}^i$.

- Given the pre-subnetwork at $\mathbf{w_s}$, we write the explicit formula for $\frac{\partial u_i}{\partial w_s}$:

$$\frac{\partial u_i}{\partial w_s} = f[w_s](X_i) \cdot \sum_{p \in pre(w_s)} \text{path\_gradient}_\text{p}(w_s, X_i) \tag{10}$$

where $\text{path\_gradient}_p(w_s, X_i)$ is the gradient value along the path $p$ at the left vertex of $w_s$, and can be expanded as follows using the chain rule:

$$\text{path\_gradient}_p(w_s, X_i) = (\tau_0^i w_0 \tau_{p_1}^i w_1 \ldots \tau_{p_{l(p)-1}}^i w_{l(p)-1})(p) \tag{11}$$

we refer to $\text{path\_gradient}_p(w_s, X_i)$ as $\text{PG}_p(w_s, X_i)$ for brevity from now on.

**Theorem 5.1.** *For a network $f$,*

$$H_{ij} = \sum_w \left( f[w](X_i) \cdot f[w](X_j) \cdot \left( \sum_{p \in pre(w)} \text{PG}_p(w, X_i) \right) \cdot \left( \sum_{p \in pre(w)} \text{PG}_p(w, X_j) \right) \right) \tag{12}$$

which easily follows by substituting Equation (10) in Equation (4). In the rest of the paper, we will make many arguments most of which rely on the number of terms in this equation. Number of terms in Equation 12 is itself controlled by the number of paths in the pre-subnetworks at each weight. Following examples provide some intuition.

**Example 5.2.** *In this example, our network has a structure as illustrated in Figure 2a. The output node is denoted with value one, indicating that the activation function at output layer is an identity function. As shown in Figure 2b, there are only one path in each weight's pre-subnetwork. Therefore, there are only three terms in Equation 12, where each summand is of form $\text{PG}_p(w, X_i)\text{PG}_p(w, X_j)$. We refer to these terms as symmetrical pairs since both multipliers are representing the same path.*

**Example 5.3.** *A more complicated application of the formula can be found in Figure 2d. There is one path in each of the following weights' presubnetwork: $w_0, w_1, w_2, w_3,$ and $w_4$, and three paths at $w_5$'s presubnetwork. Therefore, Equation 12 would have $5 + 3^2 = 14$ summands, 8 of which are terms with symmetrical pairs and 6 are cross-terms induced by the paths in $w_5$'s pre-subnetwork. In general, cross-terms will appear only if there are two paths that share a final edge which can only happen if there are three or more number of layers in the network.*

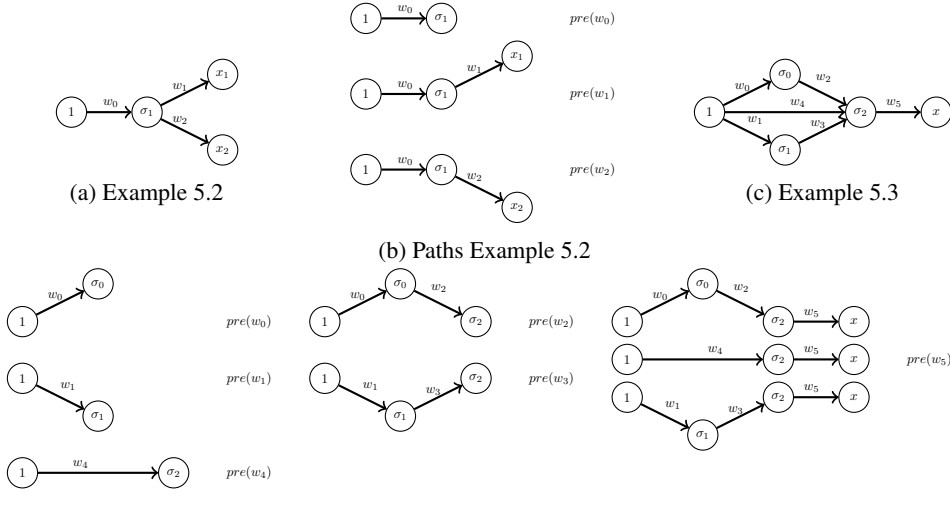

Figure 2: Path Decomposition Examples.

# 6 ARCHITECTURE IMPLICATIONS

## 6.1 CONVERGENCE RATE SCALES LINEARLY WITH WIDTH

In this Section, we show that for sufficiently wide neural networks, convergence rate has a linear relationship with the network's width. Results are drawn theoretically at time 0 assuming random i.i.d initialization. For these results to generalize across time, we assume that $H$ stays close to its initial value. In the Appendices D and E, we show empirical results that suggest this assumption is true. For the extreme case of infinitely-wide networks, it has been shown that $H$ is constant throughout training [22]. Others [5, 15] have also shown that for sufficiently-wide neural networks weights remain close to initialization, and so our argument holds true across training.

**Proposition 6.1.** *For a 2-layer neural network with hidden dimension $m$, the expected convergence rate scales linearly with $m$.*

To demonstrate the simplicity and expressiveness of our approach, next we will discuss proofs with and without our approach.

**Brute-force approach** A 2-layer fully-connected linear neural network can be modeled as:

$$u(X) = V \times W \times X = \begin{pmatrix} v_1 & \dots & v_m \end{pmatrix} \times \begin{pmatrix} w_{11} & \dots & w_{1d} \\ \vdots & & \vdots \\ w_{m1} & \dots & w_{md} \end{pmatrix} \times \begin{pmatrix} x_1 \\ \vdots \\ x_d \end{pmatrix} \quad (13)$$

where $V \in \mathbb{R}^{1,m}$ and $W \in \mathbb{R}^{m,d}$ are the weights of the second and first layer, respectively and the input $X$ is a $d$-dimensional column vector. We begin the analysis with the simple case of a 2-layer linear network where the network's width is one, i.e. $m = 1$ and $u(X) = v_0 \cdot \langle W(:,1), X\rangle$.

$H_{ij}$ can be decomposed into two terms, partial derivatives with respect to weights of the first layer and partial derivatives with respect to weights of the second layer.

$$H_{ij} = \frac{\partial u_i}{\partial v_0}\frac{\partial u_j}{\partial v_0} + \sum_{w \in W}\frac{\partial u_i}{\partial w}\frac{\partial u_j}{\partial w} = \langle W(:,1), X_i\rangle \cdot \langle W(:,1), X_j\rangle + v_0^2 \sum_{k=1}^{d} X_{i,k} \cdot X_{j,k} \quad (14)$$

For a more general case where the width is $m$, we have:

$$H_{ij} = \sum_{l=1}^{m}\left(\langle W(:,l), X_i\rangle \cdot \langle W(:,l), X_j\rangle + v_l^2 \sum_{k=1}^{d} X_{i,k} \cdot X_{j,k}\right) \quad (15)$$

Adding non-linearity would make this equation even more complex, which is the approach used in previous literature together with induction:

$$H_{ij} = \sum_{l=1}^{m} \left( \sigma(\langle W(:,l), X_i \rangle) \cdot \sigma(\langle W(:,l), X_j \rangle) + v_l^2 \sum_{k=1}^{d} X_{i,k} \cdot \sigma'(\langle W(:,l), X_i \rangle) \cdot X_{j,k} \cdot \sigma'(\langle W(:,l), X_j \rangle) \right) \tag{16}$$

See Appendix B for details of how we derive these equations. Since we assume all weights $(w_{lk}, v_0)$ are independent and identically distributed, the summands within Equations (15) and (16) are also independent and identically distributed. This indicates that $H_{ij}$ is composed of m i.i.d components, hence convergence rate scales linearly with $m$.

**Our approach: Using Theorem 5.1** More generally, by applying Theorem 5.1, it is immediate to see every pre-subnetwork will be a path in a 2-layer fully connected network. There are two types of paths; $m$ length 1 paths and $m$ length 2 paths. The weights are independent and follow the same distribution, hence, a 2-layer network's $H$ matrix always decomposes as $m$ i.i.d sums. Hence convergence rate scales linearly with $m$. This proves Proposition 6.1. □

See Section 7 for empirical analysis that motivated this study. We show for a 2-layer linear and non-linear networks that convergence rate has linear relationship with width.

## 6.2 DEPTH-INDUCED FIRST-ORDER MOMENTUM

In this section, we use our formula to explain how depth has a momentum-like effect. Take a network $f$. Construct a new network $g$ by adding a multiplication layer on top, i.e. $g(X) = \mu f(X)$ with an extra trainable parameter $\mu$. We first present an inequality (with the proof left to Appendix F), then explain why the inequality implies momentum-like behavior and finally how they are different.

**Proposition 6.2.** *With $u_g = \mu u_f$,*

$$cos(\frac{du_g}{dt}|_{u_g}, u_g) \geq cos(\frac{du_f}{dt}|_{u_f}, u_f) \tag{17}$$

*where cos denotes cosine between two vectors.*

**Remark 6.3** (Momentum interpreted as directional preference)**.** *In physics, momentum is defined as the tendency of an object to follow its current velocity of motion. In gradient based optimization, momentum can be defined as in Equation (1) of [32],*

$$v_{t+1} = m_0 v_t - \epsilon \nabla f(w_t) \tag{18}$$

*where $v$'s are the updates, $\epsilon$ is the learning rate and $m_0$ is the momentum coefficient. Quantitatively, $cos(v_{t+1}, v_t)$ increases with $m_0$. A bigger $m_0$ implies a stronger preference in $v_{t+1}$ for the direction of $v_t$. For network depth, Proposition 6.2 describes the stronger directional preference of the deeper network g in the direction of the current value vector. In the depth-induced momentum experiment below, we show that this directional preferences can have similar effects on convergence rates as momentum in gradient based optimization.*

**Remark 6.4** (Difference to momentum)**.** *The key difference between depth-induced momentum and gradient descent momentum is the following. The depth-induced momentum is a condition between $u'$ and $u$, whereas gradient descent momentum is a second order condition between $u''$ and $u'$ (see [29] Eq. (4) and (10)).*

**Remark 6.5** (Sensitivity to output scale)**.** *Since $\langle u', u \rangle = \frac{1}{2}\frac{d}{dt}\langle u, u \rangle = \frac{1}{2}\frac{d}{dt}\|u\|^2$, depth-induced momentum will be more pronounced when $\|u\|$ can increase. In Appendix G, we present an experiment that shows that depth-induced momentum is more pronounced when output scale is larger.*

## 6.3 ON THE IMPORTANCE OF NUMBER OF PATHS FOR CONVERGENCE

Here, we will provide some heuristic arguments, based on our formula, on why over-parametrization through increasing the number of paths can be more effective than increasing the number of nodes (parameters) and conclude that increasing depth is more effective than increasing width for accelerating the training process. We first introduce an important terminology which we will use below.

**Definition 6.6.** *Given a collection of vectors $\{g_i\}$, the Gram matrix $G$ is defined as a matrix whose $ij$-th entry is $\langle g_i, g_j \rangle$ under the Euclidean inner product.*

**Remark 6.7.** *Based on the definition above it is easy to see that Gram matrix is symmetric and positive-semidefinite and its rank is exactly the dimension of the subspace spanned by $\{g_i\}$. Suppose $H = H_1 + \cdots + H_n$ are all Gram matrices. Then $H$ is positive definite if any of the $H_i$ is positive definite. This is because of supperadditivity of the minimum eigenvalue function, i.e. $\lambda_{min}(H) \geq \lambda_{min}(H_1) + \cdots + \lambda_{min}(H_n)$.*

**Strongly-Gram vs. Weakly-Gram** Equation(12) in Theorem 5.1 can be expanded further as follows and the terms can be classified into two groups. The first group will include all the symmetrical pairs and the second group will include all the cross-terms.

$$H_{ij}(f) = \sum_w \left( \sum_{p \in pre(w)} \mathrm{PG}_p(w, X_i) f[w](X_i) \right) \cdot \left( \sum_{p \in pre(w_s)} \mathrm{PG}_p(w, `X_j) f[w](X_j) \right)$$

$$= \sum_w \sum_{p \in pre(w)} \mathrm{PG}_p(w, X_i) f[w](X_i) \cdot \mathrm{PG}_p(w, X_j) f[w](X_j) + \tag{19}$$

$$\sum_w \sum_{\substack{p_1, p_2 \in pre(w) \\ p_1 \neq p_2}} \mathrm{PG}_{p_1}(w, X_i) f[w](X_i) \cdot \mathrm{PG}_{p_2}(w, X_j) f[w](X_j)$$

$$= \sum_p \mathrm{PG}_p(w, X_i) f[w](X_i) \cdot \mathrm{PG}_p(w, X_j) f[w](X_j) +$$

$$\sum_w \sum_{\substack{p_1, p_2 \in pre(w) \\ p_1 \neq p_2}} \mathrm{PG}_{p_1}(w, X_i) f[w](X_i) \cdot \mathrm{PG}_{p_2}(w, X_j) f[w](X_j)$$

$$= H_{ij}^S(f) + H_{ij}^W(f) \tag{20}$$

We refer to the first group as **strongly Gram**, and denote it by $H_{ij}^S(f)$, and refer to the second group as **weakly Gram** and denote it by $H_{ij}^W(f)$, i.e. we have:

**Remark 6.8** (Gramness of the strongly-Gram component). *Suppose there are $q$ paths in the network, $p_1, p_2, \ldots, p_q$. Then Equation (19) implies that the strongly Gram component is Gramian with vectors*

$$g_i = \left( \mathrm{PG}_{p_1}(w, X_i) f[w](X_i), \quad \ldots \quad , \mathrm{PG}_{p_q}(w, X_i) f[w](X_i) \right) \tag{21}$$

**Remark 6.9** (Insignificance of the weakly Gram component). *We claim that the weakly Gram component is insignificant, see Appendix C for heuristics and empirical evidence.*

**Remark 6.10** (Number paths vs. number of nodes). *The Gram matrix $H^S$ should ideally be of full rank for good convergence. So we want the subspace spanned by $\{g_i\}$ to have dimension at least $N$. $N$ is the dimension of $H$ and also the number of sample points. On the other hand, as shown in Equation 21 $g_i$'s are embedded as vectors in a $q$-dimensional vector space where $q$ is the number of paths. Therefore, to ensure $H$ is full-rank, $q$ should satisfy $q \geq N$. The bigger $q$ is, the more likely they are linearly independent and make the matrix full rank. In deep learning, $N$ is usually large, therefore, we need to have networks whose number of paths is comparable to $N$ for best convergence.*

**Remark 6.11** (Depth vs. width). *Adding a new parameter at a new layer can increase the number of paths more effectively than adding it at an existing layer. In other words, making models deeper increases the number of paths faster than making models wider.*

# 7 EXPERIMENTS

In this section, we provide experimental results that demonstrate the impact of model structure on convergence.

**Setup** We employ a teacher-student framework to put our ideas to test. Our teacher model is a two-layer neural network with hidden dimension 100, input dimension 50 and output dimension 1. We assume a Gaussian distribution both for the inputs and ground-truth weights. The student models are also two-layer neural networks with variable hidden dimensions, and $l_2$ loss. The convergence rate at step $t$ is defined as $1 - \frac{Loss_{t+1}}{Loss_t}$.

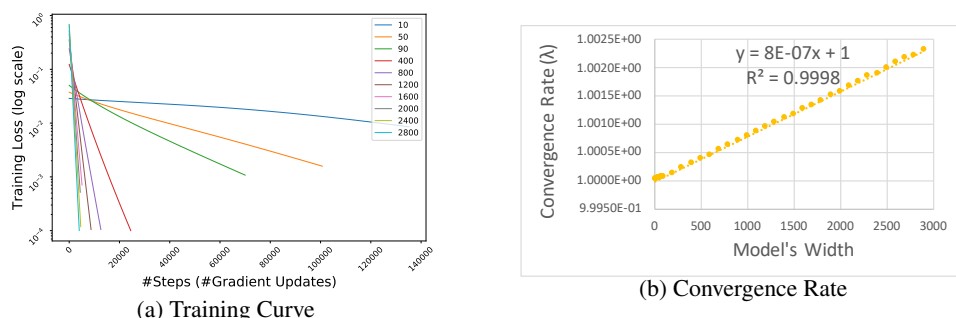

(a) Training Curve  (b) Convergence Rate

Figure 3: **Training Curve Characterization for Two-layer Linear Neural Network.**

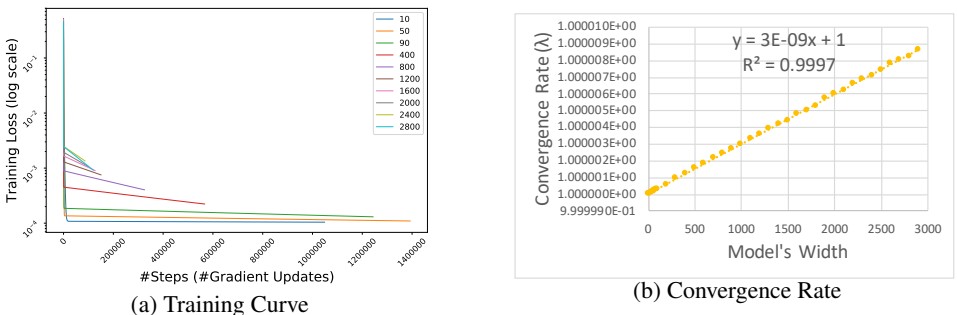

(a) Training Curve  (b) Convergence Rate

Figure 4: **Training Curve Characterization for Two-layer Non-Linear Neural Network.**

**Two-Layer Linear Neural Network** As shown in Figure 3a, training curve in log scales look like straight lines. As shown in Figure 3b, convergence rate ($\lambda$) grows linearly with width.

**Two-Layer Non-Linear Neural Network** As shown in Figure 4a, training curve for models with the sigmoid non-linearity begins in a steep region where loss drops very quickly for a short period of time and ends in a steady region where loss drops very slowly. We are interested in characterizing the curve in the steady region. As shown in Figure 4b, convergence rate ($\lambda$) grows linearly with width.

**Depth induced momentum** Figure 5 compares the convergence rate curves for 2-layer, 3-layer and 4-layer NN using GD, against a 2-layer network with a momentum optimizer. The deeper networks are constructed from 2-layer networks with additional scalar multiplication layers. As discusses in Section 4.5, one can observe depth-induced momentum in convergence rates, similar to using momentum optimizer with a 2-layer NN.

## 8 CONCLUSION

In this paper, we provide a path-based combinatorial formula for the coefficient matrix $H$ which governs the gradient descent dynamics. We advocate analyzing $H$ with this path-based approach by showing the insights we could derive from this.

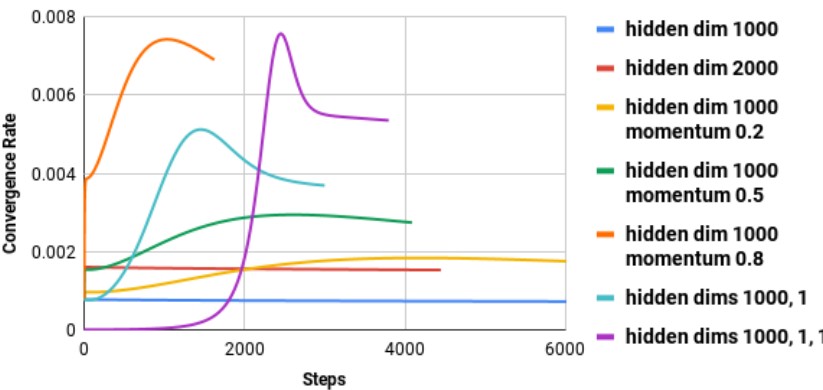

Figure 5: **Depth induced momentum.**

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

## A $l_p$ LOSS

We argue very briefly that without loss of generality, we can assume that we work with $l_2$ loss instead of other $L_p$ norms. For $l_p$ loss,

$$\frac{\partial L}{\partial u_j} = p \cdot (y_j - u_j)^{p-1} \tag{22}$$

So for $l_2$ loss, Equation 5 becomes

$$\frac{d\mathbf{u}}{dt} = \eta H(\mathbf{y} - \mathbf{u}) \tag{23}$$

Recall the argument for decay rate under Equation 8. The single variable analogue of the system is:

$$\frac{df(t)}{dt} = hf(t) \tag{24}$$

the solution looks like:

$$f(t) = c_0 e^{ht} \tag{25}$$

and decays like:

$$f(t) = c_0(1 - \xi)^t \tag{26}$$

with

$$\xi = 1 - e^h \tag{27}$$

When $h$ is small, by Taylor expansion

$$\xi \approx 1 - (1 + h + \frac{h^2}{2}...) \approx -h \tag{28}$$

Instead of using the closed form solution, this ODE could equivalently be solved using the series method and we would be getting the Taylor expansion of equation (25) on the right and equation (28) remains the same. Note that the series method (more fundamentally, Taylor expansion involving only one matrix) works for both ODEs and systems with convergent assumptions on the coefficient matrix $H$. $\eta H$ is small by assumption, so we have convergent Taylor expansions, so we continue to argue for $l_p$ loss as if we are in the single variable case.

For general $l_p$ loss, we have:

$$f' = hf^{p-1} \tag{29}$$

then

$$f = c_0 e^{\frac{h}{p}t} \tag{30}$$

and the decay rate would be $\frac{h}{p}$, scaled down by $p$, so all the convergence rate statements we make for the rest of the paper for $l_2$ remains true if scaled by $\frac{p}{2}$.

## B 2 LAYER LINEAR NETWORK

The detailed brute-force calculation of $H_{ij}$ for a 2-layer fully connected linear network with $m = 1$ is as follows. Recall $m$ is the width of the hidden layer.

$$
\begin{aligned}
H_{ij} &= \sum_{v \in V} \frac{\partial u_i}{\partial v_s} \frac{\partial u_j}{\partial v_s} + \sum_{w \in W} \frac{\partial u_i}{\partial w} \frac{\partial u_j}{\partial w} \\
&= \langle W(:,1), X_i \rangle \cdot \langle W(:,1), X_j \rangle + \sum_{k=1}^{d} (v_0 \cdot w_{1k} \cdot X_{i,k}) \cdot (v_0 \cdot w_{1k} \cdot X_{j,k}) \\
&= \langle W(:,1), X_i \rangle \cdot \langle W(:,1), X_j \rangle + v_0^2 \sum_{k=1}^{d} (w_{1k} \cdot X_{i,k}) \cdot (w_{1k} \cdot X_{j,k})
\end{aligned} \tag{31}
$$

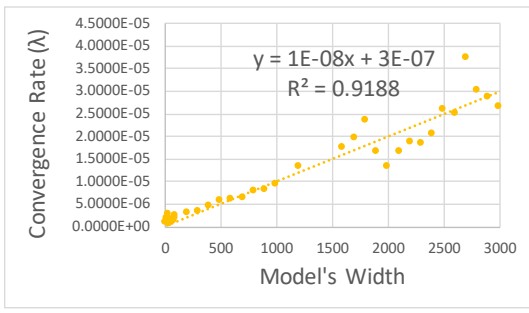

Figure 6: Convergence Rate for a 3-layer RELU-activated Network.

For a general $m$, we have:

$$
\begin{aligned}
H_{ij} &= \sum_{v \in V} \frac{\partial u_i}{\partial v_s} \frac{\partial u_j}{\partial v_s} + \sum_{w \in W} \frac{\partial u_i}{\partial w} \frac{\partial u_j}{\partial w} \\
&= \sum_{l=1}^{m} \langle W(:,l), X_i \rangle \cdot \langle W(:,l), X_j \rangle + \sum_{l=1}^{m} \sum_{k=1}^{d} (v_l \cdot X_{i,k}) \cdot (v_l \cdot X_{j,k}) \\
&= \sum_{l=1}^{m} \left( \langle W(:,l), X_i \rangle \cdot \langle W(:,l), X_j \rangle + \sum_{k=1}^{d} (v_l \cdot X_{i,k}) \cdot (v_l \cdot X_{j,k}) \right) \\
&= \sum_{l=1}^{m} \left( \langle W(:,l), X_i \rangle \cdot \langle W(:,l), X_j \rangle + v_l^2 \sum_{k=1}^{d} X_{i,k} \cdot X_{j,k} \right)
\end{aligned}
\tag{32}
$$

## C  WEAKLY GRAM MATRIX IS EMPIRICALLY INSIGNIFICANT

In terms of network architecture, the weakly Gram term would only appear when two paths share a final edge. That means the network needs to have at least 3 layers. The number of pairs could grow quadratically with width. Heuristically, each term $\mathbb{E}[\mathrm{PG}_{p_1}(w, X_i) \cdot \mathrm{PG}_{p_2}(w, X_j)]$ tend to be 0 because of independent weights in PG and the weights having expectation 0. Hence, we focus on the strongly Gram part for the rest of the section. As in Equation (19), the number of terms in weakly Gram component grows quadratically in the width of the hidden layer. One might expect this results in a quadratic growth in convergence, given all the terms are i.i.d. random variables and strictly positive. In order to test this hypothesis, we exploit a teacher-student framework as outlined in Section 7, however the student models are 3-layer RELU-activated networks to ensure the number of terms in weakly Gram component is non-zero. We choose RELU activation function to ensure that the expected value of the activation to be greater than zero. Our experimental results shows that the convergence rate relationship with width is linear (see Figure 6) and not quadratic. This implies that weakly Gram components have expectation zero.

## D  JUSTIFY $H$ STAYS CLOSE TO INITIALIZATION

Weights and $H$ are indeed time-dependent and follow their respective stochastic process. However, theoretical results in the overparametrized regime suggest that every weight vector remains close to initialization (Lemma 5.3 [5] and Lemma 3.3 [15]). We also observe that the mean and variance stay very close to initialization, $\mathcal{N}(0, 0.1)$ (Figure 7). In Figure 8, we ran the Kolmogorov-Smirnov test to compare the weight distribution against the initialization distribution. As shown, the p-values are overall stable and far from significance levels to invalidate the null hypothesis ($H_0$: trained weights follow $\mathcal{N}(0, 0.1)$).

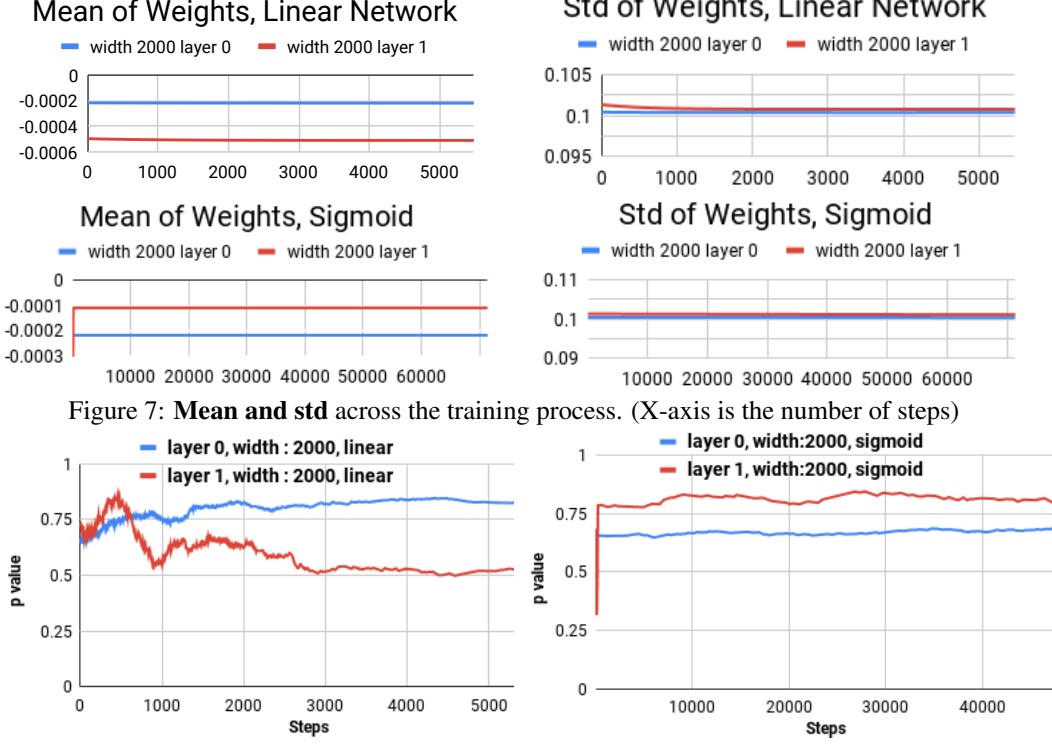

Figure 7: **Mean and std** across the training process. (X-axis is the number of steps)

Figure 8: **KS test p-values** across the training process.

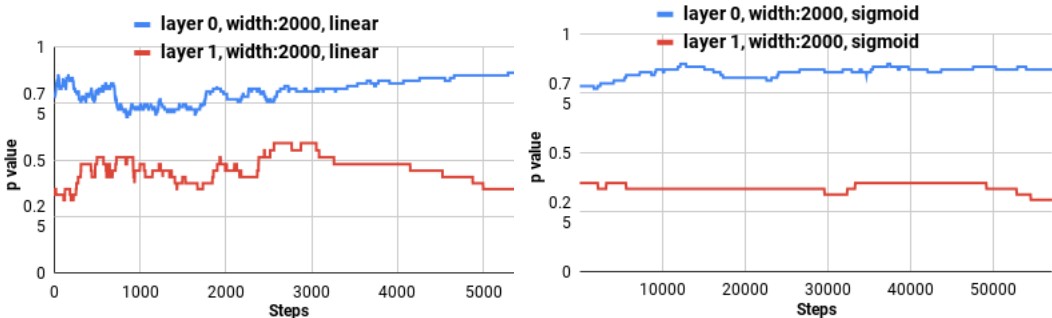

Figure 9: **Turning point test p-values**

## E    JUSTIFY IID ASSUMPTION FOR WEIGHTS

The Turning Point Test is commonly used to check iid-ness. Figure 9 shows the p-value through the training process. As shown, p-values do not exhibit any major regime shifts and the null hypothesis (the weights are iid samples) stays far from rejectable with any commonly used significance levels $(0.1, 0.05 \ldots)$. iid-ness can also be seen as a consequence of closeness to initialization in response 1. By staying close to initialization, statistical properties are preserved and the weights remain an iid sample of the original distribution.

## F    PROOF OF PROPOSITION 6.2

*Proof.* By definition of $f$ and $g$, in $g$, there's a new path of length 1 whose edge is $\mu$ and all paths in $f$ are annexed by $\mu$ at the front. Hence $H_{ij}$ for the new network $g$ can be computed in terms of $H_{ij}$

for the network $f$ as follows:

$$H_{ij}(g) = \mu^2 H_{ij}(f) + f(X_i) \cdot f(X_j) \tag{33}$$

This can be written in the matrix form as:

$$H(g) = \mu^2 H(f) + u_f \cdot u_f^T \tag{34}$$

We first calculate an inner product,

$$\langle H(g)u_g, u_g \rangle = \langle (\mu^2 H(f) + u_f \cdot u_f^T)u_g, u_g \rangle \tag{35}$$

$$= \langle (\mu^2 H(f) + u_f \cdot u_f^T)\mu u_f, \mu u_f \rangle \tag{36}$$

$$= \langle \mu^2 H(f) \cdot \mu u_f, \mu u_f \rangle + \langle u_f \cdot u_f^T \cdot \mu u_f, \mu u_f \rangle \tag{37}$$

$$= \mu^4 \langle H(f)u_f, u_f \rangle + \mu^2 \langle u_f \cdot u_f^T \cdot u_f, u_f \rangle \tag{38}$$

$$= \mu^4 \langle H(f)u_f, u_f \rangle + \mu^2 \langle \|u_f\|^2 u_f, u_f \rangle \tag{39}$$

$$= \mu^4 \langle H(f)u_f, u_f \rangle + \mu^2 \|u_f\|^4 \tag{40}$$

Now we calculate cosines,

$$cos(H(g)u_g, u_g) = \frac{\langle H(g)u_g, u_g \rangle}{\|H(g)u_g\|\|u_g\|} \tag{41}$$

$$= \frac{\mu^4 \langle H(f)u_f, u_f \rangle + \mu^2 \|u_f\|^4}{\mu^2 \|H(g)u_f\|\|u_f\|} \tag{42}$$

$$= \frac{\mu^2 \langle H(f)u_f, u_f \rangle + \|u_f\|^4}{\|H(g)u_f\|\|u_f\|} \tag{43}$$

Using equation (34) to expand the denominator,

$$\langle H(g)u_g, u_g \rangle = \frac{\mu^2 \langle H(f)u_f, u_f \rangle + \|u_f\|^4}{\|(\mu^2 H(f) + u_f u_f^T)u_f\|\|u_f\|} \tag{44}$$

$$\geq \frac{\mu^2 \langle H(f)u_f, u_f \rangle + \|u_f\|^4}{\|\mu^2 H(f)u_f\|\|u_f\| + \|u_f\|^4} \tag{45}$$

It suffices to check

$$\frac{\mu^2 \langle H(f)u_f, u_f \rangle + \|u_f\|^4}{\|\mu^2 H(f)u_f\|\|u_f\| + \|u_f\|^4} \geq \frac{\langle H(f)u_f, u_f \rangle}{\|H(f)u_f\|\|u_f\|} = \text{RHS} \tag{46}$$

which after cross-multiplication, reduces to:

$$\|H(f)u_f\|\|u_f\| \geq \langle H(f)u_f, u_f \rangle \tag{47}$$

This follows from Cauchy–Schwarz inequality. Equality holds precisely when $H(f)u_f$ is parallel to $u_f$ (Equation (47), (44)). □

**Remark F.1.** *One further remark that could be made after Remark 6.3 and 6.4 is the following. The proof above tells us that depth-induced momentum comes mainly from the extra term $u_f$ in Equation (34). Note that $u_f$s are the values of the networks at sub-network $f$. Comparing this with our path-based expansion formula at Equation (12) where all sub-networks' values appear with different scaling factors proportional to their path gradients sum, we can conclude qualitatively that this phenomenon is present across all sub-networks and each sub-network contributes to overall momentum proportional to its path gradients sums.*

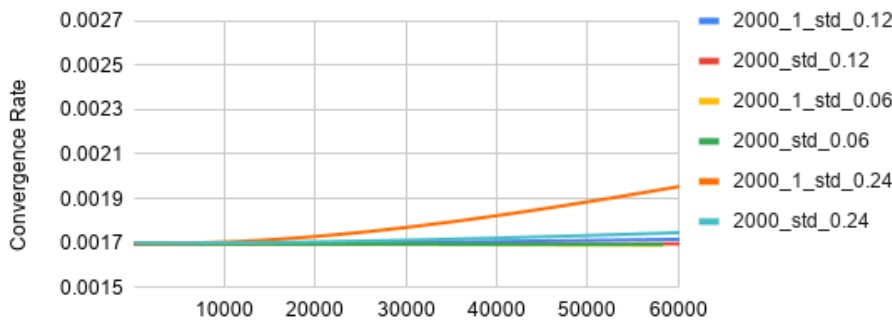

Figure 10: **Momentum's dependence on output scale.**

## G    THE IMPACT OF OUTPUT VALUE DISTRIBUTION ON DEPTH-INDUCED MOMENTUM

Figure 10 shows the impact of output distribution on momentum. The underscore separated strings represent hidden dimensions. "2000" represents hidden dimension of width 2000 and "2000_1" represents hidden layers of width 2000 and 1. The number after "std" represents the output's standard deviation in the teacher network, equivalently scale as the mean is 0. As we can observe from the graph, momentum acceleration is not present for small standard deviation (0.06 and 0.12) and the behavior is more similar to a shallow 2-layer network of the same width. When standard deviation is raised to 0.24, then we observe acceleration of the convergence rate. In all of these experiments, $\mu$ is initialized to 1 and empirically stays close to 1 to counteract the effect of $\mu^2$ in equation (34).

