# OpenReview forum: "Model Architecture Controls Gradient Descent Dynamics: A Combinatorial Path-Based Formula"
_ICLR.cc/2020/Conference — Reject_

### Official Review · AnonReviewer3 · 2019-10-23
**Official Blind Review #3**

**Rating:** 6

**Review:**

This paper presents a simple and intuitive interpretation of the dynamics of gradient descent between labels and predictions rewritten in terms of all possible paths from inputs to outputs in FC networks. The basic starting point is from the gradient flow (gradient descent in continuous time) by Du et al 2018 [13] on the difference y-u of labels y predictions u (eq (6) or (7) in the paper) instead of the standard gradient flow dW/dt = - ∂L(W)/∂w. The coefficient matrix H of the system is known to determine the convergence properties, and this can be rewritten with respect to path-wise sums of gradients through the chain rule (Theorem 4.1).  This provides some intuitive interpretations for several facts discussed in the community: 1) the linear convergence rate is more intuitively obtained compared to the naive derivations (Remark 5.2 with comparison to [13]). 2) the 'depth' of FC network affects the convergence like momentum (section 5.2). 3) the number of paths compared to the number of nodes has a predominant impact on the convergence (section 5.3). The paper also demonstrates several experiments to understand the impact of depth or paths on the convergence.

Though the paper's contribution is a quite simple path decomposition using chain rules for the coefficient matrix H of the gradient flow, it indeed provides several intuitive understandings on the impact of paths onto the gradient-descent convergence. All implications are basically confirming already known things in different (path-based) words, and the impact or novelty is rather small, but nevertheless, it would be informative.

One concern is about the description in section 4.1 and 4.2 for preliminaries results. I would suggest that emphasizing the fact 4.1 as an already discussed fact makes easy for readers to follow the results and focus more on the paper's contribution. The cited paper by Du et al [13] was published at the last year's ICLR, and the contents in 4.1 and 4.2 would be due to it including the formulation as gradient flow (gradient descent in continuous time). Given that it is already known that H is the determining factor for the convergence, then the contribution and usefulness of the proposed new representation of eq (15) should be discussed. At the first glance, the use of gradient flow w.r.t prediction u instead of parameter w might be (misleadingly) seen as the idea of this paper. The sentences in the abstract can also mislead readers to this. Also check the paper by Du et al in ICML 2019 'Gradient descent finds global minimum of deep neural networks'.

Two minor points:
1) parentheses for citing references and eq numbers are quite confusing. Please check the style format for citations.
2) in Figure 2, pre(w_0) is written as "presum(w_0)".


**Experience Assessment:**

I have read many papers in this area.

**Review Assessment: Checking Correctness Of Derivations And Theory:**

I assessed the sensibility of the derivations and theory.

**Review Assessment: Checking Correctness Of Experiments:**

I assessed the sensibility of the experiments.

**Review Assessment: Thoroughness In Paper Reading:**

I read the paper thoroughly.

---

> ### Author Response · Authors · 2019-11-14
> **Responses to Reviewer #3**
>
> One concern is about the description in section 4.1 and 4.2 for preliminaries results. I would suggest that emphasizing the fact 4.1 as an already discussed fact makes easy for readers to follow the results and focus more on the paper's contribution. The cited paper by Du et al [13] was published at the last year's ICLR, and the contents in 4.1 and 4.2 would be due to it including the formulation as gradient flow (gradient descent in continuous time). Given that it is already known that H is the determining factor for the convergence, then the contribution and usefulness of the proposed new representation of eq (15) should be discussed.
> Thanks for the suggestion. The hope is that through symmetry, the path abstraction hides away the book keeping of summations and indices. The uses we came up with are now in a separate Section 6 which separately discusses the easy way to see expected convergence rate scale linearly with width in 6.1. More exact analysis than previous literature of depth and momentum in 6.2 and path count is important for convergence in 6.3. The remark F.1 may also be of interest as a network-wide perspective.
>
>
> At the first glance, the use of gradient flow w.r.t prediction u instead of parameter w might be (misleadingly) seen as the idea of this paper. The sentences in the abstract can also mislead readers to this. Also check the paper by Du et al in ICML 2019 'Gradient descent finds global minimum of deep neural networks'.
>
> Sorry for the confusion. We moved all of prior work into a separate section and give references to Gradient descent finds global minimum of deep neural networks as [14].
>
>
> Two minor points:
> 1) parentheses for citing references and eq numbers are quite confusing. Please check the style format for citations.
> Fixed
> 2) in Figure 2, pre(w_0) is written as "presum(w_0)".
> Fixed.

---

### Official Review · AnonReviewer1 · 2019-10-23
**Official Blind Review #1**

**Rating:** 3

**Review:**

This paper studies the training dynamics of a neural network model as a dynamical system.  The authors proposed a path-based approach to compute the derivatives that would appear in the H matrix which governs the learning dynamics.  They further utilized this formulation to (1) simplify the analysis of convergence rate of 2-layer neural networks w.r.t. width; and (2) presented an argument for the similarity between added depth in the network and momentum-based optimization; and (3) also argued about the importance of the number of paths for fast convergence.

The dynamical systems view of the learning process is quite new to me, even though I’m aware of a few recent papers exploring this.  I found the paper to be mostly clear and not too hard to follow, and the dynamical systems perspective interesting.  The path gradient is quite intuitive, but I’m a bit surprised there’s no prior work (at least not discussed in this paper) studying the relationship between gradients and the paths in the network.

It is a bit hard for me to judge the significance of this work because of my lack of context.  The main implications of the path gradients were presented in section 5.  The first part shows that using their theorem 4.1 can simplify the derivation of the linear relationship between convergence rate and the width of 2-layer nets.  This is a simplification but the original derivation is not complicated either.  The second part tried to draw a relationship between added depth in a network and momentum-based optimization, which I found to be a bit hand-wavy.  In particular, I found the argument that, du/dt prefers the direction of u itself which carries information about past directions therefore this is similar to momentum, to be not convincing.  The third implication argues that we need H to be full-rank for fast convergence, and in order to achieve this we want to increase the number of paths in the network, which is interesting.

There are a few other things that could be clarified:
- the terminology in section 4.3 is a bit confusing, w_s seems to be associated with an edge in the graph, but is also referred to as a node, “we denote the activation value at node w_s with activation(w_s, X_i) ”.
- on page 6 there is a reference to Equation 5.2 which does not exist
- the results in Figure 4a is counter-intuitive - is this showing wider networks actually converge slower?  Isn’t this against the argument of this paper?
- it is unclear how the convergence rate lambda values are computed in Figure 4b, the curves in Figure 4a clearly doesn’t follow an exponential decay pattern.

Overall I found this paper presented some interesting ideas, but may need a bit more work to be ready to be published.  Happy to change my judgement however, if other more experienced reviewers can comment better on the significance of this work.

**Experience Assessment:**

I do not know much about this area.

**Review Assessment: Checking Correctness Of Derivations And Theory:**

I carefully checked the derivations and theory.

**Review Assessment: Checking Correctness Of Experiments:**

I assessed the sensibility of the experiments.

**Review Assessment: Thoroughness In Paper Reading:**

I read the paper thoroughly.

---

> ### Author Response · Authors · 2019-11-14
> **Responses to Reviewer #1**
>
> In particular, I found the argument that, du/dt prefers the direction of u itself which carries information about past directions therefore this is similar to momentum, to be not convincing.
>
> We updated the exposition of this to be the new Section 6.2. We replaced the words with an inequality which hopefully makes the point more explicit.
>
>
> The third implication argues that we need H to be full-rank for fast convergence, and in order to achieve this we want to increase the number of paths in the network, which is interesting.
>
> Thank you for liking the point.
>
>
> There are a few other things that could be clarified:
> - the terminology in section 4.3 is a bit confusing, w_s seems to be associated with an edge in the graph, but is also referred to as a node, “we denote the activation value at node w_s with activation(w_s, X_i) ”.
>
> Thanks for pointing this out. We say the value at the right vertex of w_s in the next sentence but we meant the node after the edge w_s here.
>
>
> - on page 6 there is a reference to Equation 5.2 which does not exist
>
> This is fixed with a better exposition. Please take a look.
>
> - the results in Figure 4a is counter-intuitive - is this showing wider networks actually converge slower?  Isn’t this against the argument of this paper?
>
> We unified our definition of convergence rate. We mean $1-\frac{Loss_{t+1}}{Loss_{t}}$.
> The figure is drawn in log scale. So convergence rate is the slope of the lines (after an initial “shock” that we can’t quite explain). Wider networks have steeper lines hence bigger convergence rate.
>
>
> - it is unclear how the convergence rate lambda values are computed in Figure 4b, the curves in Figure 4a clearly doesn’t follow an exponential decay pattern.
>
> The convergence rates are taken as the slopes of the lines after the initial “shock” and plotted against model width. We should have stated $1-\frac{Loss_{t+1}}{Loss_{t}}$ whereas the exponential decay only fits figure 3.

---

### Official Review · AnonReviewer2 · 2019-10-27
**Official Blind Review #2**

**Rating:** 3

**Review:**

This paper considers the problem of understanding the impact of deep neural networks (DNN) model architecture on the convergence rate of gradient descent dynamics. To achieve this goal, the paper follows the recent trend of continuous-time perspective of optimization, and proposes to model gradient descent via the gradient flow, which is a first-order ODE. The induced loss dynamics is then also following a first-order ODE with a coefficient matrix H (that depends on the solution trajectory, and hence non-constant). The paper then claims that the convergence rate is characterized by the minimum eigenvalue of H, and analyzes this H through a straightforward path-based formula obtained by chain rules. In particular, the authors try to explain the effect of width, depth and number of paths on the convergence rate, and validated these through a few numerical experiments.

Admittedly, the idea of this paper is interesting. However, I think the novelty and rigorousness of this paper is not convincing, as explained in more details below.

1. On the novelty side, characterizing the convergence via the H matrix is not new, and most of the discussions in Section 4 have appeared in exactly the same form in the previous works [13] (two-layer) and [25] (general), which are also cited in this paper. In addition, the path formula is also very closely related (if not completely the same as) with the expansion of G matrix in Section 4 of [25] (where G is the H in this paper), which decomposes the G (or H) matrix into summation over layers. These facts largely lower the contribution of this paper on a high level.

2. On the rigorousness side, the paper is not very consistent in the notation.
1) The notation is not very consistent. The authors use H in the notation section to denote the matrix, use X_i to denote the data input, but use \bf{H} and x_i (lower cased) subsequently.
2) In Section 4, the authors immediately start with the continuous-time perspective, without even mentioning the gradient dynamics or some related ansatz. The authors may want to mention that they use the ansatz w_k=W(kh), where W is a smooth curve, and take h to 0 to obtain the ODE models, as is done in [1].
3) The notation list at the beginning of Section 4.3 is too long and not clear. In particular, l(p) is not even defined before appearing in (14), and \sigma_s seems to be overridden by the notation activation(w_s, X_i) and does not appear later in the path gradient, which is weird. Shouldn't there be \sigma_s inside the formula of (14)?

3. Again on the rigorousness side, the paper is very non-rigorous when stating the claims and theorems.
1) The authors claim that "This result will hold for other l_p losses", but indeed what holds is different for l_p losses (the rate is scaled by p/2).
2) Section 4.2 does not make any sense. The authors should either directly cite the corresponding content in [25], which are much clearer, or directly invoke the standard linear ODE theory and use the matrix exponential and Taylor expansion to make the explanations.
3) The paper seems to use a very informal argument that H stays close to its initial value to derive all the theory, which is only empirically checked in the appendix. But given the proportion of the theory part of this paper, I think the paper should either clearly state the assumption as H being constant, or follow the manner of [25] to prove that H stays close to some fixed matrix and use this to prove the other theorems rigorously. Otherwise, the theory part is both hard to understand and verify.
4) Proposition 5.1 should clearly state whether the statement is in the expectation sense, or high probability sense, or something else.
5) The explanation in Section 5.2 is rather unclear. In particular, I don't understand why "eigenvalues of H(g) are pushed in the direction of g" implies that "the updates prefer the direction of u", and why this then further implies something related to the momentum acceleration. The authors should provide a rigorous statement here.
6) The authors claim at the bottom of page 1 that they show adding a new layer leads to H(t) being decomposed into an adaptive learning rate term and a momentum term. But the adaptive learning rate part is not showing anywhere later in the paper.

Minor comments:
1) In equation (5), there should be a minus sign on the right-hand side.
2) In Section 5.2, there is no "Equation 5.2". It should be something else.
3) The authors may want to add citations to [2] (which is a concurrent work with [25] on essentially the same topic) and [3] (which is a predecessor work of neural ODE).

[1] Su, Weijie, Stephen Boyd, and Emmanuel Candes. "A differential equation for modeling Nesterov’s accelerated gradient method: Theory and insights." Advances in Neural Information Processing Systems. 2014.
[2] Allen-Zhu, Zeyuan, Yuanzhi Li, and Zhao Song. "A convergence theory for deep learning via over-parameterization." arXiv preprint arXiv:1811.03962 (2018).
[3] Lu, Yiping, et al. "Beyond finite layer neural networks: Bridging deep architectures and numerical differential equations." arXiv preprint arXiv:1710.10121 (2017).

################## post rebuttal ##################
After reading the authors' rebuttal, I decide to raise my rating to 3 (weak reject).

**Experience Assessment:**

I have read many papers in this area.

**Review Assessment: Checking Correctness Of Derivations And Theory:**

I assessed the sensibility of the derivations and theory.

**Review Assessment: Checking Correctness Of Experiments:**

I assessed the sensibility of the experiments.

**Review Assessment: Thoroughness In Paper Reading:**

I read the paper thoroughly.

---

> ### Author Response · Authors · 2019-11-14
> **Response to Reviewer #2 editorial questions**
>
> 2. On the rigorousness side, the paper is not very consistent in the notation.
> 1) The notation is not very consistent. The authors use H in the notation section to denote the matrix, use X_i to denote the data input, but use \bf{H} and x_i (lower cased) subsequently.
> We made H consistent and uses $X_i$ as a vector and $x_i$ a number consistently now.
>
> 2) In Section 4, the authors immediately start with the continuous-time perspective, without even mentioning the gradient dynamics or some related ansatz. The authors may want to mention that they use the ansatz w_k=W(kh), where W is a smooth curve, and take h to 0 to obtain the ODE models, as is done in [1].
> We fixed this by adding the reference for going from step-wise gradient descent to the continuous time gradient flow at the top of Section 4.
>
> 3) The notation list at the beginning of Section 4.3 is too long and not clear.  In particular, l(p) is not even defined before appearing in (14),
>
> l(p) refers to the length of the path p. We clarified this in the new version and made sure that the concepts are sequentially introduced.
>
> and \sigma_s seems to be overridden by the notation activation(w_s, X_i) and does not appear later in the path gradient, which is weird. Shouldn't there be \sigma_s inside the formula of (14)?
>
> We removed the notation activation(w_s, X_i). By \sigma_s, we are guessing the reviewer meant \omega_s? So \omega_s is not part of the product because in the end, the path gradient is part of the derivative with respect to \omega_s, so \omega_s won’t be part of it. path_gradient(w, X_i) is more like w.r.t. w more than a function of w.
>
> 3. Again on the rigorousness side, the paper is very non-rigorous when stating the claims and theorems.
> 1) The authors claim that "This result will hold for other l_p losses", but indeed what holds is different for l_p losses (the rate is scaled by p/2).
> We did point out the constant scale in the appendix. We also added “up to a constant” in the main text.
>
> 2) Section 4.2 does not make any sense. The authors should either directly cite the corresponding content in [25], which are much clearer, or directly invoke the standard linear ODE theory and use the matrix exponential and Taylor expansion to make the explanations.
> 	As part of moving the old Section 4.1, 4.2 into a separate background Section. We removed this subsection, only kept the conclusion and refer to [25] now at the beginning of the section.
>
> Minor comments:
> 1) In equation (5), there should be a minus sign on the right-hand side.
> Fixed.
> 2) In Section 5.2, there is no "Equation 5.2". It should be something else.
> Reorganized.
> 3) The authors may want to add citations to [2] (which is a concurrent work with [25] on essentially the same topic) and [3] (which is a predecessor work of neural ODE).
> Thanks for pointing these out. References added.

---

> ### Author Response · Authors · 2019-11-14
> **Response to Reviewer #2**
>
> 1. On the novelty side, characterizing the convergence via the H matrix is not new, and most of the discussions in Section 4 have appeared in exactly the same form in the previous works [13] (two-layer) and [25] (general), which are also cited in this paper. In addition, the path formula is also very closely related (if not completely the same as) with the expansion of G matrix in Section 4 of [25] (where G is the H in this paper), which decomposes the G (or H) matrix into summation over layers. These facts largely lower the contribution of this paper on a high level.
>
> They are closely related and they calculate essentially the same object. Previous work used layer by layer recursive expressions. The path expansion is novel. The convergence rate and width argument is used to promote this exploitation of path symmetry more to lighten the burden of book keeping for further research. For the remaining two contributions, the path expansion is used to gain further insights into the relationship between depth and momentum in value space and path counts and convergence.
>
> 3. Again on the rigorousness side, the paper is very non-rigorous when stating the claims and theorems.
> 3) The paper seems to use a very informal argument that H stays close to its initial value to derive all the theory, which is only empirically checked in the appendix. But given the proportion of the theory part of this paper, I think the paper should either clearly state the assumption as H being constant, or follow the manner of [25] to prove that H stays close to some fixed matrix and use this to prove the other theorems rigorously. Otherwise, the theory part is both hard to understand and verify.
>
> Our work is built on ideas from prior work that shows (1) for sufficiently large neuraL network H stays close to its initial value. (2) In overparametrized models the weights stay close to their initial value, hence H remains close to its initial value. We have clarified this in our updated version.
>
> 4) Proposition 5.1 should clearly state whether the statement is in the expectation sense, or high probability sense, or something else.
>
> Thanks for pointing this out. It is in the expectation sense, so we modified it to say the expected convergence rate.
>
> 5) The explanation in Section 5.2 is rather unclear. In particular, I don't understand why "eigenvalues of H(g) are pushed in the direction of g" implies that "the updates prefer the direction of u", and why this then further implies something related to the momentum acceleration. The authors should provide a rigorous statement here.
>
> We apologize for the exposition of the old Section 5.2. It is now Section 6.2. We provided the rigorous statement as an inequality and discussed the similarities and differences between this momentum acceleration and momentum acceleration used in gradient descent. Please take a look.
>
> 6) The authors claim at the bottom of page 1 that they show adding a new layer leads to H(t) being decomposed into an adaptive learning rate term and a momentum term. But the adaptive learning rate part is not showing anywhere later in the paper.
>
> Adaptive learning rate is captured by $\mu^2$ of Equation (21) in original submission and in Equation (34) in the updated version. In the new revision, we focus more on the discussion of momentum in the forefront and adaptive learning rate only appears in the appendix F now.

---

### Author Response · Authors · 2019-11-14
**Submission update and response to common questions**

We thank the reviewers for the helpful feedback and interests in various points made in the paper.  On a high level, we made the following changes. Each individual comment will be addressed below. We reorganized the old Section 4.1 and Section 4.2 into a separate Section called background to make clearer the separation between prior work and our contribution. We also thank reviewers for the feedback on the writing of the subsection on momentum. We re-formalized the conclusion with a Proposition then give remarks on similarities and differences. We think this would be a much clearer exposition.

We hope all these changes help distinguish our novelty and contribution better, which can be summarized as:

(1) We show that H can be expressed in terms of the number of paths and the form of the paths. To the best of our knowledge, this has never been shown before. The closest to our work is [25] (now [14]) which studies H inductively over layers (not a global summation over paths) and model it recursively (not directly). After all H can be expanded in many different ways but how to expand it gives us the power to analyze the  impact of model architecture on convergence.
(2) Showing that expected convergence rate scales linearly with network width. This demonstrates the path-based decomposition exploits the networks’ path symmetry better compared to the recursive expressions. The hope is that this formalism liberates future researchers from working with many indices and help them reason about different activation functions universally. The conclusion also focuses on the expected convergence rate while varying network width instead of convergence bounds at the limit as prior works do.
(3) We characterize the depth acceleration effect quantitatively, study the similarities and differences between depth-induced momentum and the commonly applied momentum in gradient descent. We observe some effects only observable through the value space analysis. All of this is done under less restrictive assumptions on the network. This study of momentum in value space is new.
(4) We show that the number of paths compared to the number of nodes has a predominant impact on the convergence using our global path based expansion and that the Gram matrix H needs to be full-rank for fast convergence, and in order to achieve this, we want to increase the number of paths in the network.

---

### Decision · Program_Chairs · 2019-12-19

**Decision:**

Reject

**Comment:**

This paper focuses on understanding the role of model architecture on convergence behavior and in particular on the speed of training. The authors study the gradient flow of training via studying an ODE's coefficient matrix H. They study the effect of H in terms of possible paths in the network. The reviewers all agreed that characterizing the behavior in terms of path is nice. However, they had concerns about novelty with respect to existing work on NTK. Other comments by reviewers include (1) poor literature review (2) subpar exposition and (3) hand-wavy and rack of rigor in some results. While some of these concerns were alleviated during the discussion. Reviewers were not fully satisfied.  I general agree with the overall assessment of the reviewers. The paper has some interesting ideas but suffers from lack of clarity and rigor. Therefore, I can not recommend acceptance in the current form.